

# Water-level attenuation in broad-scale assessments of exposure to coastal flooding: a sensitivity analysis

Athanasios T. Vafeidis[1], Mark Schuerch[2], Claudia Wolff [1], Tom Spencer[2], Jan L. Merkens[1], Jochen Hinkel[3], Daniel Lincke[3], Sally Brown [4], Robert J. Nicholls [4]

[1]*Coastal Risks and Sea-Level Rise Research Group, Department of Geography, Christian-Albrecths University Kiel, Ludewig-Meyn-Str. 14, 24098 Kiel, Germany*
[2]*Cambridge Coastal Research Unit, Department of Geography, University of Cambridge, Downing Place, Cambridge CB2 3EN, UK*
[3]*Global Climate Forum e.V. (GCF), Neue Promenade 6, 10178 Berlin, Germany*
[4]*Faculty of Engineering and the Environment, University of Southampton, Highfield, Southampton SO17 1BJ, UK*

*Correspondence to*: Athanasios T. Vafeidis (vafeidis@geographie.uni-kiel.de)

**Abstract.** This study explores the uncertainty introduced in global assessments of coastal flood exposure and risk by not accounting for water level attenuation due to land-surface characteristics. We implement a range of plausible water level attenuation values in the flood module of the Dynamic Interactive Vulnerability Assessment (DIVA) modelling framework and assess the sensitivity of flood exposure and flood risk indicators to differences in attenuation rates. Results show a reduction of up to 47% in area exposure and even larger reductions in population exposure and expected flood damages. Despite the use of a spatially constant rate for water attenuation the reductions vary by country, reflecting the differences in the physical characteristics of the floodplain as well as in the spatial distribution of people and assets in coastal regions. We find that uncertainties related to the omission of this factor in global assessments of flood risk are of similar magnitude to the uncertainties related to the amount of SLR expected over the 21[st] century. Despite using simplified assumptions, as the process of water level attenuation depends on numerous factors and their complex interactions, our results strongly suggest that future impact modelling needs to focus on an improved representation of the temporal and spatial variation of water levels across floodplains by incorporating the effects of relevant processes.

## 1. Introduction

Increased flooding due to sea-level rise (SLR) is the primary natural hazard that coastal regions will face in the 21[st] century, with potentially high socio-economic impacts (Kron, 2013; Wong et al., 2014). Broad-scale (i.e. continental to global) assessments of coastal flood exposure and risk are therefore required in order to inform mitigation targets and adaptation decisions (Ward et al., 2013a), related financial needs and loss and damage estimates. Towards these ends, a number of recent studies have assessed the exposure of area, population and assets to coastal flooding at national to global scales (Nicholls, 2004; Brown et al. 2013; Jongman et al., 2012a; Ward et al., 2013b; Arkema et al., 2013) as well as flood risk (Hinkel et al. 2014).



Although the methods for broad-scale coastal-flood exposure and risk assessment vary between studies, flood extent and water
depth are usually assessed based on spatial analysis, assuming that all areas with an elevation below a certain water level that
are hydrologically connected to the sea are flooded (the "bathtub" method) (Poulter and Halpin, 2008; Lichter et al., 2010).
An exception is the study of Dasgupta et al. (2009) who used a simple approach to account for wave height attenuation with
distance from the coast. The use of simplified methods for assessing flooding is primarily related to difficulties of using
hydrodynamic methods at broad scales. These difficulties are due to the limited availability and large volume of the necessary
input data, as well as to the prohibitive computational costs, which render these approaches impractical at global scales
(Ramirez et al., 2016). Usually, global applications have utilised elevation data with a spatial resolution of 1km and a vertical
resolution of 1m (Mondal and Tatem, 2012; Jongman et al., 2012b; Ward et al., 2014), with only a few recent studies employing
higher spatial resolution (90m) datasets (e.g. Hinkel et al., 2014; see also de Moel et al., 2015).
Hydrodynamic models are normally used only for local-scale applications. This is because they require detailed data on
parameters such as coastal topography/bathymetry and land use in order to represent local-scale processes and to account for
hydraulic properties. These data are, however, not always readily available and are associated with large data volumes. A range
of simpler inundation models that partly account for hydraulic processes at intermediate scales using medium resolution
elevation data (<100m$^2$) have also been applied at subnational scales (e.g., Bates et al., 2010; Wadey et al., 2012; Lewis et al.
2015; Ramirez et al., 2016), and these models are beginning to inform analysis at even broader scales (e.g., Vousdoukas et al.,
2016). There is also a developing literature on hydrodynamic modelling of water level attenuation over coastal wetlands at the
landscape scale (<1km) for saltmarshes (Loder et al., 2009; Wamsley et al., 2009, 2010; Barbier et al., 2013; Smith et al.,
2016) and mangrove forests (McIvor et al., 2012; Zhang et al., 2012; Liu et al., 2013). However, the incorporation of the above
processes in global models is still very limited.
Not accounting for hydrodynamic processes in global models can lead to overestimation of flood extent and water depth.
Hydrodynamic models capture processes that are not included in global models, e.g. the effects of surface roughness (both
natural and anthropogenic) and channel network density and connectivity (and its effect on landscape continuity) on the timing,
duration and routing of floodwaters. For example, inundation extent has been shown in some cases to significantly decrease
in urban and residential areas when the built environment is represented in numerical simulations (e.g. tsunami inundation:
Kaiser et al., 2011; storm surge inundation: Brown et al., 2007; Orton et al., 2015).
To our knowledge there is no study that has explored the uncertainty introduced into global models by not accounting for water
level attenuation due to hydrodynamic processes related to surface roughness. This paper aims to address this gap. We derive
a range of plausible water-level attenuation values from existing literature and implement them in the flood module of the
Dynamic Interactive Vulnerability Assessment (DIVA) modelling framework (described in Hinkel et al., 2014). Next, we
assess the sensitivity of flood exposure and flood risk indicators to plausible changes in water-level attenuation values under



a range of different SLR scenarios. Finally, we compare the uncertainty due to water level attenuation rates with the uncertainty range associated with expected SLR during the 21$^{st}$ Century.

## 2. Methods and Data

### 2.1 The Dynamic Interactive Vulnerability Assessment (DIVA) modelling framework

DIVA is an integrated, global modelling framework for assessing the biophysical and socio-economic consequences of SLR, and associated extreme water levels, under different physical and socio-economic scenarios and considering various adaptation strategies (Hinkel and Klein, 2009). DIVA has been widely used for global and continental scale assessments of SLR impacts, vulnerability and adaptation (e.g., McLeod et al., 2010; Hinkel et al. 2010; Brown et al. 2013; Hinkel et al., 2013; Hinkel et al., 2014; Spencer et al., 2016). It is underpinned by a global coastal database which divides the world's coastline (excluding Antarctica) into 12,148 coastal segments (Vafeidis et al., 2008). Each segment contains approximately 100 elements of data concerning the physical, ecological and socio-economic characteristics of the coast. For the purposes of the present study, we focus on the impacts of increased exposure to coastal flooding and potential damages of extreme sea level events (due to the combination of storm surges and astronomical high tides). We used the flood algorithm of DIVA (for details see Hinkel et al., 2014) to estimate potential coastal flood damage, SLR impacts and associated costs.

We specifically considered the following five indicators, which progressively include additional components of flood risk:

1. Area below the 1-in-100 year flood event (km$^2$), an estimate based on elevation data and information on water levels for a single hazard event (i.e. the height of the 1-in-100 year sea flood);

2. People living in the 1-in-100 year floodplain, a calculation based on spatial data on elevation and population as well as on information for a single hazard event (i.e. the height of the 1-in-100 year sea flood);

3. Assets in the 1-in-100 year floodplain (US $), a calculation that uses data on elevation, population, Gross Domestic Product (GDP) and information for a single hazard event (i.e. the height of the 1-in-100 year sea flood);

4. Expected value of the number of people flooded per year (hereafter, people flooded), a calculation based on elevation and population data and the probability distribution of the hazard (i.e. sea flood heights and their probability of occurrence); and

5. Expected value of annual damages to assets (hereafter, flood damage) (US $), a calculation based on elevation, population and GDP data and the probability distribution of the hazard (i.e. sea flood heights and their probability of occurrence).

For each coastline segment, a cumulative exposure function for area and population that gives the areal extent (hydrologically connected to the sea) and number of people below a given elevation was constructed. Damages to assets were assessed using a depth-damage function with a declining slope, with 50% of the assets being destroyed at a water depth of one metre (Messner et al., 2007).



### 2.2 Coastal Elevation and Rate of Water level Attenuation

To simulate the effect of different values of attenuation at the broad scale, we implemented a stylised elevation profile in order to represent the process of water level attenuation. We assumed that water levels decrease at a constant slope ($\alpha$) with increasing distance from the coastline. Location-specific coastal profiles for every coastline segment were based on floodplain areas contained within the DIVA database. The database reports total land area within different elevation increments (<1.5m, 1.5-2.5m, 2.5-3.5m, 3.5-4.5m, 4.5-5.5m, 5.5-8.5m, 8.5-12.5m, 12.5-16.5m) for each coastal segment. The elevation dataset that was used for estimating floodplain areas and developing the segment elevation profiles is the Shuttle Radar Terrain Mission (SRTM) Digital Elevation Database (Jarvis et al., 2008) which has a vertical resolution of 1m and a spatial resolution of 3 arc seconds (~90m at the equator).

We approximated the average coastal profile for every segment by assuming that elevation continuously increases with distance from the shore. Starting with the lowest elevation increment, the floodplain areas of all elevation increments were cumulatively summed to retrieve the total area below a certain elevation. The total areas are then divided by the segment length to derive the inundation length of the respective floodplain ($dx_i$).

To evaluate the representativeness of the assumption of continuously increasing elevation with increasing distance from the shore, we used the original SRTM dataset and calculated the Euclidian distance of each cell to the nearest coastline for every pixel. Mean distances from the coast were calculated for each of the floodplain areas of each segment. Subsequently, we compared these mean distances with the respective average floodplain elevation for each DIVA coastline segment to analyse the validity of the "continuous-increase" assumption. This comparison revealed that 55% of the DIVA coastline segments show either a continuous increase or no change in the mean distance along the elevation profile (Figure 1a), suggesting that elevation does not decrease with distance from the coast. Comparing all elevation increments of all segments (i.e. pairwise comparison of the mean distances of consecutive elevation increments in a segment), there was an increase, or no change, in the mean distance from the coastline in 88% of cases. Only 12% of cases showed a decrease (Figure 1b). This result indicates that the stylised continuous profile (Figure 1a) can be regarded as generally representative of global coastal topography.

(a)                                                      (b)

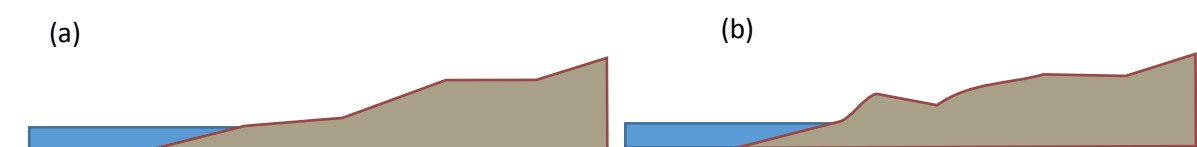

**Figure 1: Stylised coastal profile with (a) continuous and (b) discontinuous increase in elevation with distance from the shore.**

We then adjusted the coastal profile using a range of possible attenuation rates that represent different water surface slopes. Depending on the applied value for water level attenuation, the slope ($\alpha$) of the inundating water surface was employed to



modify (incline) the coastal profile. Based on this slope, the coastal profile is thereby elevated by the amount of the water level
reduction ($hx_i$) computed at a distance $dx_i$ (Fig. 1):
$$hx_i = tan(\alpha) * dx_i \qquad \text{(equation 1)}$$
In this way the original floodplain areas and inundation depths are reduced in order to account for the reduced (i) inundation
length ($dx$) and (ii) inundation depth ($hx$) (see Fig. 2).

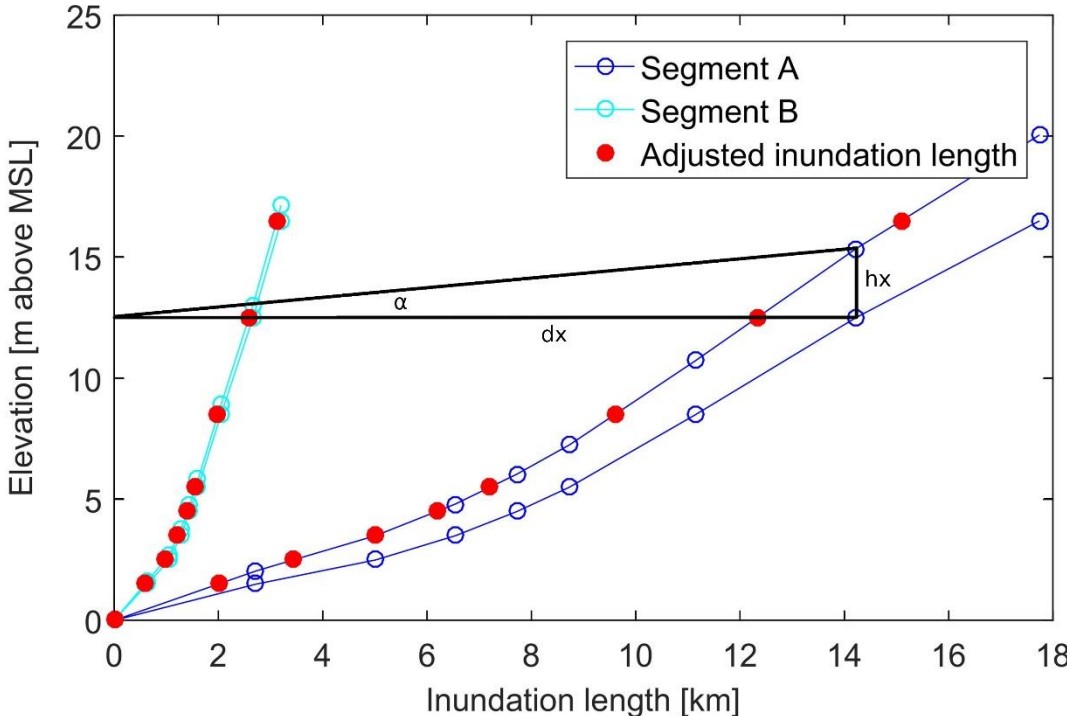


**Figure 2: The stylised coastal profile, based on the floodplain areas in the DIVA database (lower line), for two characteristic**
**coastline segments (A with a flat and B with a steep profile). Water level attenuation is accounted for by elevating the coastal**
**profile according to equation 1 (upper line). Red dots on the adjusted coastal profile indicate the inundation length in the case of a**
**water level with a constant slope of α, which represents the attenuation rate and for an incident water level equal to the**
**corresponding increment height.**
For the sensitivity analysis we used the following range of attenuation rates for inclining the water surface in order to represent
a constant water level attenuation and the associated reduction in water levels ($\alpha$): 0, 10, 20, 50 and, following discussion with
experts in the field, 100cm/km. This range embraces the values reported in the literature (Table 1), where water level under
storm conditions has been shown to decrease with distance from the coast. For reviewing the literature we employed the ISI
Web of Knowledge and based our search on the keywords "surge", "attenuation", "water-level". We selected studies that



directly reported values of water level reduction with distance and did not include studies focussing on wave attenuation. We
must note that the aim was not to carry out a systematic literature review but rather to identify a range of values that could be
used for our sensitivity analysis. The identified studies all relate to coastal wetland environments; although there are published
studies of localised water level dynamics from flow-form interactions in urban and other settings, we have not come across
similar landscape-scale assessments for other landuse types. Therefore we extended this review, where reported attenuation
values were up to 70cm/km, by directly contacting experts with experience in field or modelling studies. Following their
suggestions we decided to investigate attenuation rates of up to 100 cm/km as an upper limit.

| Event type | Landcover type | Location | Rate of water-level reduction | Method | Source |
|---|---|---|---|---|---|
| Storm surge | Bare land and Marsh | Modelled platform +0.5 m above sea level | 10 cm / km (no vegetation, no channels)<br><br>26 cm / km (100% vegetation cover, no channels)<br><br>8 cm / km (100% vegetation cover, channel network) | Numerical modelling | Temmerman et al., 2012 |
| Hurricane Isaac (2012) | Marsh | Louisiana | Up to 70cm/km water level reduction in presence of vegetation; 37 % reduction of total inundation volume | Numerical modelling | Hu et al., 2015 |
| Hurricanes | Marsh | Multiple | 1 m per 14.5 km 6.9 cm/km<br><br>(range from 1m per 5km to 1m per 60km 20 - 1.7 cm/km) | Field Study | Corps of Engineers (1963) – In Wamsley et al., 2010 |
| Hurricane Andrew (1992) | Marsh | Louisiana | 1m per 20km-23.5km 5 - 4.3 cm/km | Field Study | Lovelace 1994 |
| Hurricane Rita (2005) | | Louisiana | 1m per 4km to 1m per 25km 25 – 4 cm/km | Field Study | McGee et al. 2006 in Wamsley et al., 2010 |
| Hurricanes | Mangroves Marsh | Florida | 9.4 - 4.2 -cm/km | Field Study | Krauss et al., 2009 |



| | | | | | |
|---|---|---|---|---|---|
| Wilma (2005) and Charley (2004) | | | | | |
| Hurricanes | Mangroves | Louisiana | 23.3 – 1.7 cm/km | Field Studies | McIvor et al., 2012 (from various studies) |
| Hurricane Wilma (2005) | Mangroves | South Florida | Up to 50 cm/km (6-10 cm per km in the absence of mangroves) | Field study & modelling | Zhang et al., 2012 |
| Hurricanes | Mangroves | South Florida | 7.7 - 5.0 cm/km | Modelling | Liu et al., 2013 |

**Table 1: Water level reduction rates, for different types of landcover, as reported in the literature.**

**2.4 Sea-Level Rise and Socio-Economic Scenarios**

For global SLR in 2100 from a 1985 – 2005 baseline, we used three scenarios: the 5% quantile of the low Representative Concentration Pathway (RCP) 2.6; the median of the medium scenario RCP 4.5; and the 95% quartile of the high scenario RCP 8.5. These scenarios are represented by SLR estimates of 29, 50 and 110 cm (by 2100), respectively and were developed in the Inter Sectoral Model Intercomparison Project Fast Track (for full details see Hinkel et al., 2014). Once mean sea level is determined, future extreme water levels are obtained by displacing upwards extreme water levels for different return periods (as included in the DIVA database) with the rising sea level, following what has been observed to occur by Menendez and Woodworth (2010).

We used a single shared socio-economic pathway (SSP), namely SSP2, to represent changes in coastal population and assets. SSP2 reflects a world with medium assumptions between the other four SSPs, in terms of resource intensity and fuel dependency as well as GDP and population development (O'Neil et al., 2014). Finally, we ran the DIVA model using a no-dike scenario, where no defence measures for preventing coastal flooding are present.

**3. Results**

We present results for five different attenuation rates, across the five indicators that progressively include additional components of flood risk.



### 3.1 Reduction of current flood exposure and risk

Table 2 shows the results from the "bath tub" model (0 cm/km) and both the absolute and percentage reductions in the values

of the five indicators against this baseline.

| Water Level Attenuation Rate | 0 cm/km | 10 cm/km (% reduction) | 20 cm/km (% reduction) | 50 cm/km (% reduction) | 100 cm/km (% reduction) |
|---|---|---|---|---|---|
| Area below the 1-in-100 year flood [km²] | 621,721 | 520,423 (16.3%) | 473,044 (23.9%) | 395,525 (36.4%) | 328,661 (47.1%) |
| People below the 1-in-100 year flood [million] | 159 | 101 (36.4%) | 85 (46.1%) | 69 (56.3%) | 58 (63.0%) |
| Assets below the 1-in-100 year flood [billion US$] | 5,920 | 4,177 (29.4%) | 3,692 (37.6%) | 3,099 (47.7%) | 2,651 (55.2%) |
| People flooded [million/yr] | 124 | 78 (37.3%) | 67 (46.2%) | 54 (56.6%) | 45 (63.3%) |
| Flood damages to assets for the 1-in-100 year flood [billion US$/yr] | 2,987 | 2,145 (28.2%) | 1,906 (36.2%) | 1,594 (46.6%) | 1,372 (54.1%) |

**Table 2: Reduction, relative to the bathtub method, of five indicators of global exposure and risk, for different water-level attenuation rates. Values are for a medium SLR scenario (median of the medium scenario RCP 4.5; 50 cm by 2100)**

Our results show that the inclusion of constant water-level attenuation rates in the assessment of flooding results in large

differences in the values of the five indicators. For example, the area exposed to the 100-year flood in 2015 reduces by up to

47% with the use of different attenuation rates. A rate of 10cm/km, which has been assigned to non-vegetated surfaces, results

in an area reduction of 16% while a rate of 50cm/km, which has been measured in mangroves, can result in a reduction of 36%

(see Table 2). Interestingly, the number of people in the 100-year floodplain reduces to 58 million for an attenuation rate of

100 cm/km. This is a reduction of 63%, which is larger than the respective reduction in assets (55%) and in area (47%). This

result reflects the high population density near the coast that has been reported in previous studies (Neumann et al., 2015).

Flood damages from the 1-in-100 year event are reduced in similar proportion, totalling a reduction of more than 1.4 trillion

US$ globally, for an attenuation rate of 100 cm/km.





Despite using a global value for the attenuation rate for every model run, the reduction in impacts is not uniform across the
globe and can vary considerably between different countries. Some examples are given in Table 3, where accounting for water
level attenuation can reduce area exposure by up to 77% in China, 57% in Bangladesh and 56% in the USA. At the same time,
the reduction in annual flood costs follows a different trend, with exposed assets reducing by up to 71% in China, 49% in
Bangladesh and 25% in the USA, reflecting differences in the physical characteristics of the floodplain as well as in the spatial
distribution of people and assets in the coastal regions of these three countries.

| Water Level Attenuation Rate | 0 cm/km | 10 cm/km (reduction) | 20 cm/km (reduction) | 50 cm/km (reduction) | 100 cm/km (reduction) |
|---|---|---|---|---|---|
| **Area below 1-in-100 year flood** (km$^2$) | | | | | |
| Bangladesh | 7723.38 | 6006.7 | 5374.86 | 4291.14 | 3326.18 |
| | | (22%) | (30%) | (44%) | (57%) |
| China | 48002.08 | 29240.27 | 22532.22 | 14971.79 | 10999.83 |
| | | (39%) | (53%) | (69%) | (77%) |
| USA | 42354.83 | 34984.95 | 30595.83 | 23870.76 | 18846.31 |
| | | (17%) | (28%) | (44%) | (56%) |
| **Assets below 1-in-100 year flood** (billion $US) | | | | | |
| Bangladesh | 58 | 43 | 39 | 34 | 31 |
| | | (25%) | (32%) | (41%) | (49%) |
| China | 2244 | 1260 | 1021 | 794 | 642 |
| | | (43%) | (54%) | (64%) | (71%) |
| USA | 267 | 250 | 240 | 221 | 200 |
| | | (6%) | (10%) | (17%) | (25%) |

**Table 3: Absolute and relative reduction of the 1-in-100-year floodplain area and associated exposed assets when applying different**
**water-level attenuation rates for Bangladesh, China and USA. Values assume a medium SLR scenario (median of the medium**
**scenario RCP 4.5; 50 cm in 2015).**

**3.2 Comparison of attenuation rate uncertainty with sea-level rise uncertainty**
Figure 3 illustrates the area of land located below the 1-in-100 year storm surge level (H100), plotted against the different
attenuation rates for water level change. The inclusion of linear water-level attenuation in the assessment of flooding results


in large differences in the calculation of area exposure for all SLR scenarios. The extent of the 100-year floodplain in 2100
(Figure 3) reduces substantially under all SLR scenarios. This reduction amounts up to 36% and 46% of the total exposed area
for the medium SLR scenario (median of the medium scenario RCP 4.5; 50 cm by 2100), and water level attenuation rates of
50cm/km and 100 cm/km respectively. The relative reduction is marginally smaller for the high SLR scenario compared to the
medium-, low- and no-SLR scenarios. Importantly, the overall difference in the extent of the area of the 100-year floodplain
between the no- and high-SLR scenarios is smaller than the difference in area extent between the 0 and 20cm/km water level
attenuation rates under any scenario. This indicates that when assessing area exposure, accounting for even relatively moderate
rates of water level attenuation can be of equal importance to the differences that result from different scenarios of SLR. This
analysis, therefore, strongly suggests that uncertainties related to the omission of this factor in global assessments of flood risk
are of similar magnitude to the uncertainties related to the magnitude of SLR expected over the 21st century.

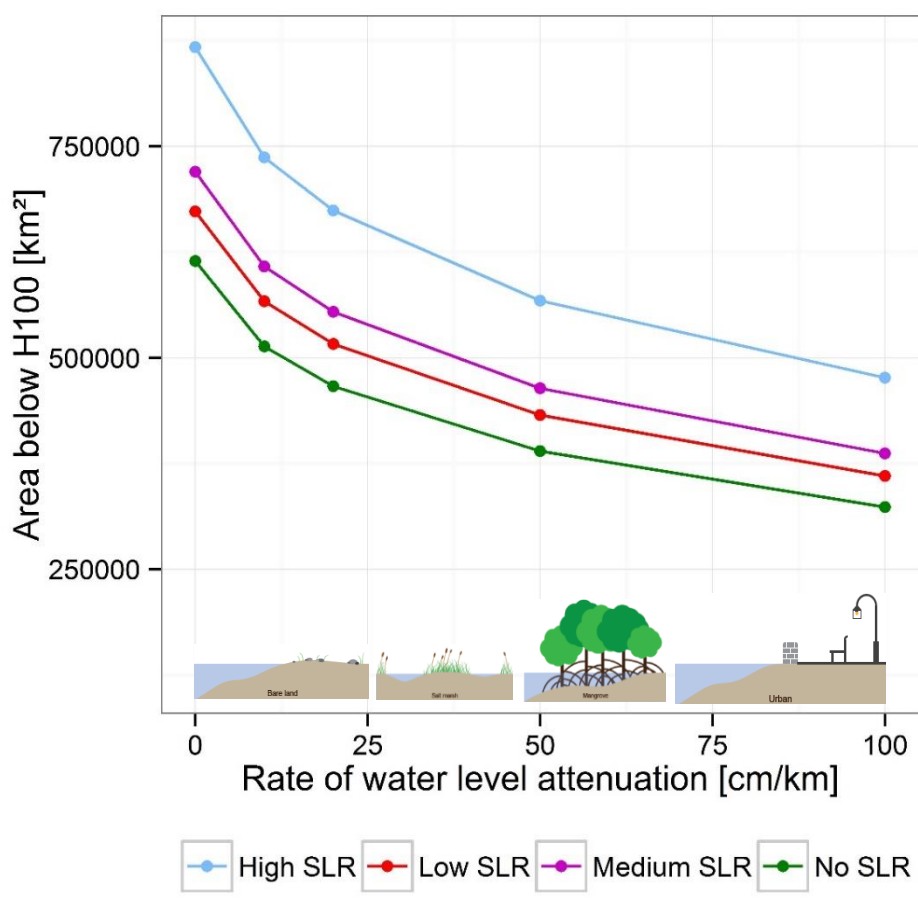


**Figure 3: Global total extent of the 100-year floodplain, for different water level attenuation rates and SLR scenarios.**



An attenuation of water levels of 10 cm/km, a typical value reported in wetlands (Table 1), results in a reduction of 100,000
km$^2$ of area exposed to the 1-in-100-year flood under the no-SLR scenario. This increases to 130,000 km$^2$ under the high SLR
scenario. It is also noteworthy that this, rather moderate, attenuation rate can shift impacts (in terms of area exposure) in time
by approximately 30 years (under all SLR scenarios).
Similar patterns can be observed for the exposure of population to the 1-in-100-year flood (Figure 4). An attenuation rate of
50cm/km, a value that has been reported in mangrove forests (Table 1), leads to a reduction of more than 50% in the exposure
of population in 2100, under the high SLR scenario, bringing the number of people at risk in the 100-year floodplain down by
136 million. Moreover, an attenuation of 20cm/km leads to a reduction in risk to 106 million people, making population
exposure lower than the exposure under no SLR when attenuation is not considered. Again, this result suggests that accounting
for water level attenuation may be equally important to accounting for SLR uncertainty when assessing the exposure of people
to coastal flooding due to SLR.

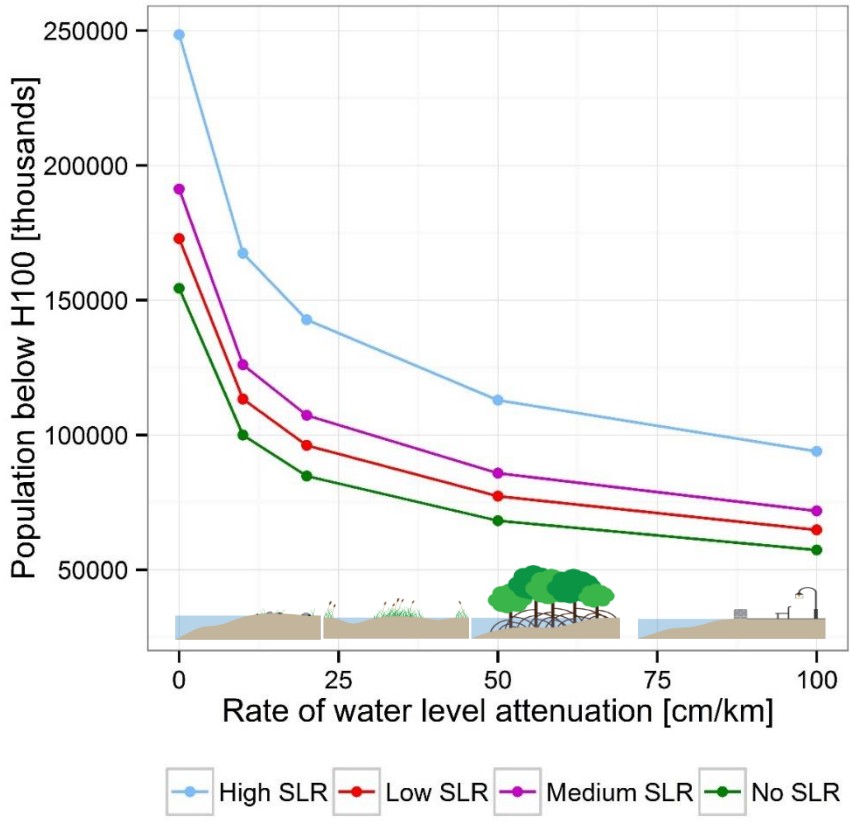


**Figure 4: Global estimates of population in the 100-year floodplain for different water-level reduction rates (Table 1) and SLR**
**scenarios.**



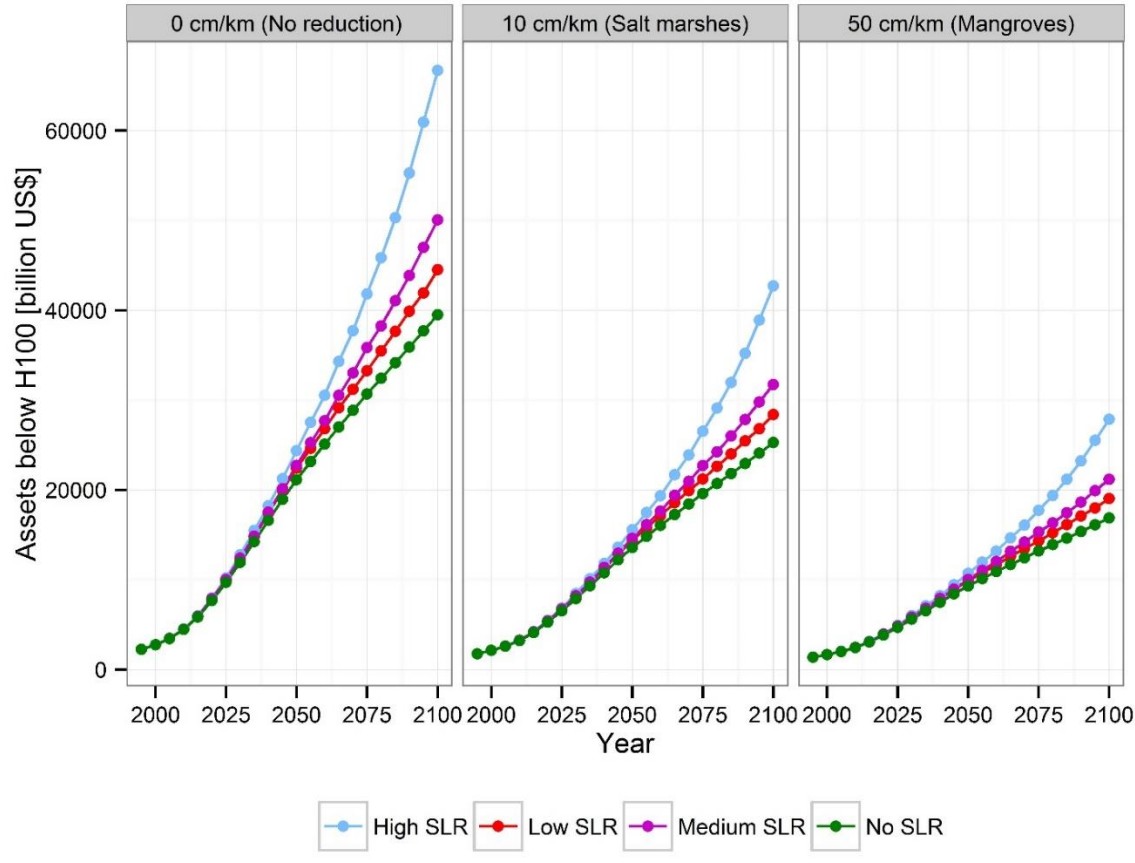

**Figure 5: Temporal evolution of the amount of assets that are located in the 100-year floodplain for different water-level reduction rates (Table 1) and SLR scenarios.**

Similar trends can be seen in the value of assets exposed to the 1-in-100-year flood, under all scenarios, when accounting for water level attenuation (Figure 5). A constant attenuation of 20cm/km would result in a decrease in the exposure of assets of approximately 37% in 2100, for a medium SLR scenario, whereas reductions of up to 48% can be seen in assets exposure when an attenuation rate of 50 cm/km is used. At the same time, damages also reduce considerably by the introduction of water level attenuation rates (Figure 6). For example, the use of an attenuation rate of 10 cm/km results in a 28% reduction in damages to assets in 2100 for the 1-in-100 year flood. The larger decrease of assets' exposure due to water level attenuation compared to population and area exposure is due to the fact that, besides the decrease of the flood area extent, water level attenuation leads to an additional reduction of flood depth with distance from the coast. As water depth is an important parameter for calculating damages to assets (Thieken et al., 2005; Penning-Rowsell et al., 2013), depth reduction further reduces the exposure of assets due to flooding.



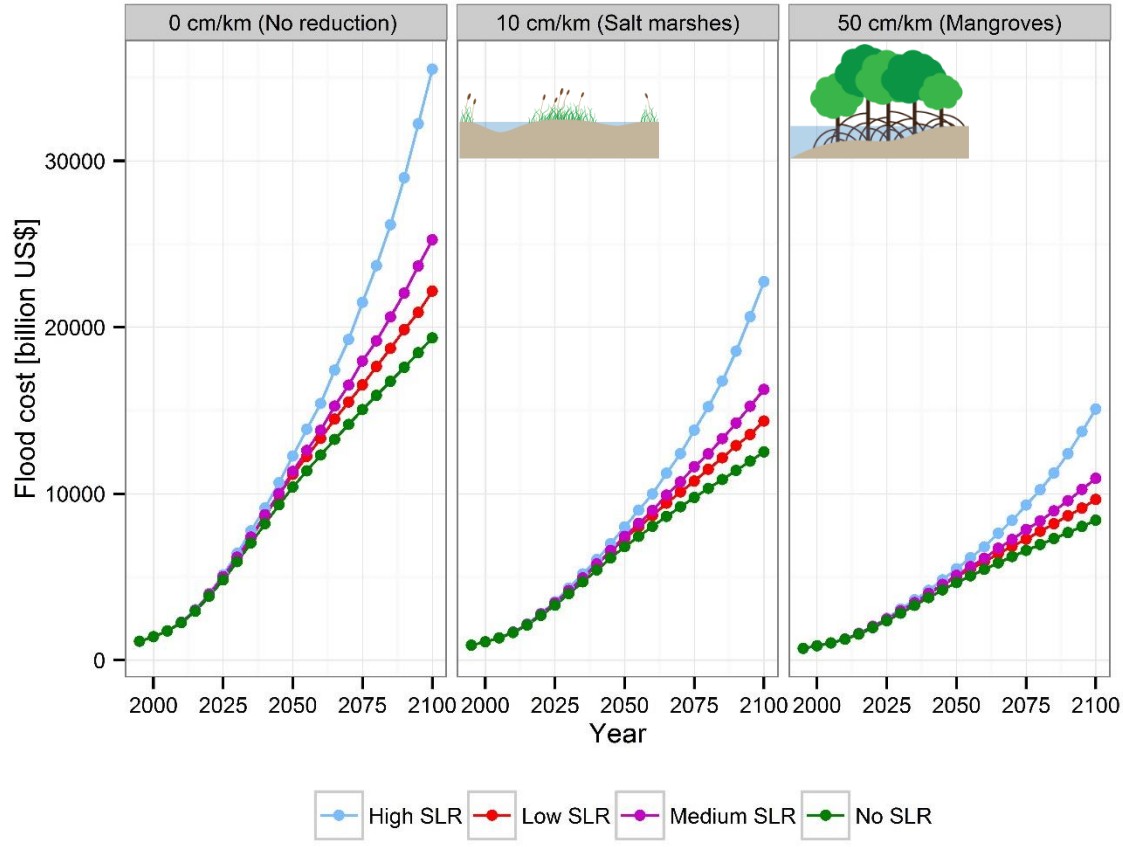

**Figure 6: Comparison of temporal evolution of sea-flood damage estimates for attenuation rates of 0, 10 and 50 cm/km, for different SLR scenarios.**

## 4. Discussion and Conclusions

The results of this study highlight the importance of accounting for the effects of hydrodynamic processes when assessing the impacts of coastal flooding at national to global scales. In particular, water level attenuation from the interaction of extreme inundation events with vegetated surfaces can lead to considerably lower estimates of exposure of land area and population to coastal flooding. Furthermore, this effect can lead to large reductions in potential damages, as lower water depths combined with smaller flood extents give significantly lower flood-damage costs. The reduction in exposure and risk is especially pronounced for water level attenuation rates of up to 20cm/km, rates which are typical of tidal wetlands.





Importantly, the effects of accounting for water level attenuation appear to be as important in assessing impacts as accounting
for uncertainties related to the total magnitude of SLR that is commonly employed for impact analyses. In many of the cases
that we explore, the difference in impacts between the no- and high-SLR scenarios is lower than the difference in impacts
between a no-attenuation and the attenuation scenario of 10 cm/km. This is of relevance in environments where the floodplain
substantially extends inland, such as in many of the world's deltas and coastal plains.
It is widely acknowledged that the use of simplified methods, such as the bathtub method, can provide useful first-order
estimates of global impacts of SLR and associated flooding. Although an overestimation of flood extent and depth with the
use of the bathtub method should be generally anticipated, the reduction that we observe with the use of water level attenuation
rates does not necessarily reflect the actual impacts. These are likely to depend on additional factors which are usually not
considered in global assessments. For example, damage to assets is based solely on water depth. Factors such as high local
flow velocities from channelized flow, storm wave impacts, inundation by saline water and sedimentation from flood waters
are not taken into account. Such contributory factors can lead to an increased cost of damages and thus counteract the lower
impacts predicted from the use of a water level attenuation term alone. Furthermore, the analysis reported here is predicated
on the assumption of a continuous increase in elevation with increasing distance from the shore. This study shows that whilst
this assumption holds true for the majority of coastal segments, there are segments where this assumption does not hold true.
In these cases model outputs may poorly describe flood areas, flooded population numbers and asset damages and incorrectly
predict the effect of changes in the rate of water level attenuation. Nevertheless, and despite these caveats, our results emphasise
the importance of accounting for uncertainties in impact assessments stemming from an inadequate consideration of water
level attenuation over coastal plains.
Our approach means to provide an illustration of the potential effects of water level attenuation, as this process is not constant
throughout the floodplain and depends on numerous parameters beyond the type of the surface cover. These factors include
storm duration, wind direction, water depth and vegetation traits (Resio and Westerink, 2008; Smith et al., 2016; Stark et al.,
2016). Furthermore, applying a constant slope to account for water level attenuation is a strong simplification, since this will
vary between different storm events, but also under the influence of SLR. Nevertheless, given the very high sensitivity of the
outputs to small changes in water level reduction and the obvious lack of sufficient data on the actual effect of different types
of surface on attenuating water levels during surges, we suggest that future work needs to focus on quantifying the water level
attenuation terms for different land uses. Thus, for example, both Brown et al. (2007), in the case of modelled flooding
following storm surge-induced sea defence failure, and Kaiser et al. (2011), in the case of modelled tsunami wave impacts,
have shown that disregarding buildings and associated infrastructure (roads, gardens, ditches) when assessing inundation can
lead to a large overestimation of the extent of flooding. Furthermore, given the large range of uncertainty with respect to the
actual values of water level reduction associated with just one surface cover, wetland habitat (Table 1), future impact modelling
needs to focus on a better understanding of the temporal and spatial variation of water levels across floodplains showing a
wide variety of landuse types and human occupancy, including densely urbanised regions (e.g. Lewis et al., 2013; Blumberg



et al., 2015). This work should include, but also go beyond, quantifying the water level reduction of coastal wetlands in order to enable broad-scale models to incorporate, initially in a stylised manner, the effects of water level attenuation.

Given that coastal wetlands can efficiently attenuate surge water levels, the results of this study give a first estimate of how much of an impact reduction may result from the implementation of large-scale, ecosystem-based flood risk reduction management schemes (e.g. Temmerman et al., 2013). In addition, achieving lower water levels through the establishment of coastal wetlands not only reduces impacts but may also affect the timing of potential adaptation tipping points by extending the anticipated lifetime of adaptation measures. This would allow the development of alternative adaptation pathways, a sequential series of linked adaptation options triggered by changes in external conditions (Barbier, 2015), for coastal regions.

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
