# Peer review of "Nat. Hazards Earth Syst. Sci. Discuss., https://doi.org/10.5194/nhess-2017-199 Manuscript under review for journal Nat. Hazards Earth Syst. Sci. Discussion started: 7 September 2017"

_Natural Hazards and Earth System Sciences, 2017_

## Referee Comment (RC1) · Anonymous Referee #1 · 4 Oct 2017

I have just finished reviewing the paper and I can provide my comments. The paper is very well written and the topic is definitely hot and of interest to a broader audience. However, I am afraid that this is where my positive comments about the manuscript end. The authors use DIVA, a large-scale coastal impact assessment tool to assess the effect of landward water level reduction to the estimated impacts. The addition of this feature to DIVA is definitely an improvement for the tool per se, but I struggle to find how the work brings new knowledge to the community; which should be the case when a new paper is published. The authors report that the estimated damages are strongly affected when the water level reduction is considered, but this is absolutely nothing new. Apart from the fact that common sense is sufficient to reach to the same conclusion there are several previous works proving that (Breilh et al., 2013; Ramirez et al., 2016; Seenath et al., 2016; Vousdoukas et al., 2016). All the above studies show how the static approach overestimates flood extents and some even demonstrate how this affects estimates of number of people affected. Among the above papers there are also large-scale applications (i.e. European scale), which could be the only 'new' element introduced from the manuscript. Even the reduction rates considered constitute no contribution since (i) they have been published elsewhere (ii) and no testing/validating takes place at least to provide some recommendations about suitable estimates to the community. At this stage it is important to highlight that the way the authors deal with coastal flooding is perfectly identical in terms of implementation to a case of inland flooding. The authors just use a DEM and forcing water levels along the coast which is identical to what the inland flooding community does with dynamic models long time ago. There are quite a few global river flooding impact assessments which use dynamic models and demonstrate that the time to leave the bathtub approach has arrived long time ago(Alfieri et al., 2017 and references therein; Winsemius et al., 2016). So starting from the point that all hydrologists and coastal engineers/geomorphologists know that bathtub has the only advantage of being simple in its implementation, the question is what could be done better. Even neglecting all the large-scale implementations of hydrological models, and considering that a simplified approach would be the way to go, the authors seem to neglect basic principles as well as all the progress recently done by the community. - All the arguments about the continuously increasing profiles related to Figure 1 are wrong. Figure 1b is much closer to what most beach profiles look like in nature than Figure 1a. Beaches have dunes, berms, dykes seawalls, etc which are not resolved by a 100 m DEM, but that doesn't make them non-existent. I consider that this point only is enough to reject the work since it raises serious concerns about the methodology followed. - The authors resolve to searching the literature for reported water level reduction rates while the answer is in the basic textbooks. Flooding intensity is attenuated by bottom friction and the community has been estimating Manning friction coefficients depending on land use class for decades. There are also text book

examples on how to relate the land use classes and the water level reduction they drive. The Flood Index Approach is based on the same principle and is very computationally light (Dottori et al., 2016). - Breilh 2013 did an exhaustive comparison of inundation approaches and proposed among others the Volume Integration approach which respects the volume of water contributing to the flood event, something the present effort omits. - As mentioned above, all the discussion until now relates to coastal flooding exercises which in terms of implementation are similar to river flooding ones. Most particularities of coastal flooding are omitted; i.e the effect of waves manifested as wave setup, runup and overwash (Matias et al., 2008; McCall et al., 2010), as well as the interaction between storm surge generation, wave propagation and flooding (Bertin et al., 2014). Projects like MICORE and RiscKit have introduced a lot of new developments on that direction which at least should be mentioned.

Given all the above I have serious concerns about the work being published at this form. The approach is outdated, the paper brings no new knowledge and there is no validation to support any claims. The authors should at least demonstrate that their approach is valid based on some kind of validation against observations. They could also bring new knowledge to the community by recommending testing such an approach and testing against others, highlighting in which cases it could be valid, proposing calibration parameters etc

References Alfieri, L., Bisselink, B., Dottori, F., Naumann, G., de Roo, A., Salamon, P., Wyser, K., Feyen, L., 2017. Global projections of river flood risk in a warmer world. Earth's Future 5, 171-182. Bertin, X., Li, K., Roland, A., Zhang, Y.J., Breilh, J.F., Chaumillon, E., 2014. A modeling-based analysis of the flooding associated with Xynthia, central Bay of Biscay. Coastal Eng. 94, 80-89. Breilh, J.F., Chaumillon, E., Bertin, X., Gravelle, M., 2013. Assessment of static flood modeling techniques: application to contrasting marshes flooded during Xynthia (western France). Nat. Hazards Earth Syst. Sci. 13, 1595-1612. Dottori, F., Martina, M.L.V., Figueiredo, R., 2016. A methodology for flood susceptibility and vulnerability analysis in complex flood scenarios. Journal of

[Figure]

Flood Risk Management, n/a-n/a. Matias, A., Ferreira, Ó., Vila-Concejo, A., Garcia, T., Dias, J.A., 2008. Classification of washover dynamics in barrier islands. Geomorphology 97, 655-674. McCall, R.T., Van Thiel de Vries, J.S.M., Plant, N.G., Van Dongeren, A.R., Roelvink, J.A., Thompson, D.M., Reniers, A.J.H.M., 2010. Two-dimensional time dependent hurricane overwash and erosion modeling at Santa Rosa Island. Coastal Eng. 57, 668-683. Ramirez, J.A., Lichter, M., Coulthard, T.J., Skinner, C., 2016. Hyperresolution mapping of regional storm surge and tide flooding: comparison of static and dynamic models. Nat. Hazards 82, 571-590. Seenath, A., Wilson, M., Miller, K., 2016. Hydrodynamic versus GIS modelling for coastal flood vulnerability assessment: Which is better for guiding coastal management? Ocean Coast. Manag. 120, 99-109. Vousdoukas, M.I., Voukouvalas, E., Mentaschi, L., Dottori, F., Giardino, A., Bouziotas, D., Bianchi, A., Salamon, P., Feyen, L., 2016. Developments in large-scale coastal flood hazard mapping. Natural Hazards and Earth System Science 16, 1841-1853. Winsemius, H.C., Aerts, J.C.J.H., van Beek, L.P.H., Bierkens, M.F.P., Bouwman, A., Jongman, B., Kwadijk, J.C.J., Ligtvoet, W., Lucas, P.L., van Vuuren, D.P., Ward, P.J., 2016. Global drivers of future river flood risk. Nature Clim. Change 6, 381-385.

---

## Author Comment (AC1) · 12 Oct 2017

We appreciate the effort and comments of the anonymous reviewer upon which we were able to realize that we have not presented our research question and results in a sufficiently clear manner. We have now improved the clarity of our presentation, specifically with respect to highlighting the novelty of our approach and results. Below, we respond to each of the points raised (reviewer's comments are included in quotation marks):

"The authors report that the estimated damages are strongly affected when the water level reduction is considered, but this is absolutely nothing new. Apart from the fact

that common sense is sufficient to reach to the same conclusion there are several previous works proving that (Breilh et al., 2013; Ramirez et al., 2016; Seenath et al., 2016; Vousdoukas et al., 2016). All the above studies show how the static approach overestimates flood extents and some even demonstrate how this affects estimates of number of people affected. Among the above papers there are also large-scale applications (i.e. European scale), which could be the only 'new' element introduced from the manuscript."

We agree with the reviewer that proving that damage estimates are strongly affected, when the water level reduction is considered, is nothing new and is, to some extent, common sense. In our paper however, we quantify this effect within global sea-level rise impact assessments, which has never been done before. The papers that the reviewer cites above (some of which are also cited in our manuscript) have not addressed this issue. Specifically:

- the cited studies do not assess flood damage, they primarily focus on the flood hazard and/or model single events

- none of the studies cited are global

- none of the studies include a range of physical and socio-economic scenarios (with the exception of Seenath et al. 2016 who only use two sea-level scenarios)

Our paper addresses those points by performing a sensitivity analysis using the range of published water-level reduction factors. Our findings are new as no study has demonstrated, among others, that the range of uncertainty due to the omission of this factor in global scale impact assessments is larger than that introduced by sea-level rise scenarios; that this omission can affect the spatial patterns and distribution of impacts; and that the inclusion of this factor can lead to a shift in the timing of impacts.

"All the arguments about the continuously increasing profiles related to Figure 1 are wrong. Figure 1b is much closer to what most beach profiles look like in nature than

Figure 1a. Beaches have dunes, berms, dykes seawalls, etc which are not resolved by a 100 m DEM, but that doesn't make them non-existent. I consider that this point only is enough to reject the work since it raises serious concerns about the methodology followed."

In principle, this point is well taken, but we disagree with the conclusions. Generally, this point relates to an old debate between local scale and global scale modellers and judging from the reviewer's comment, the reviewer probably belongs to the former camp. The essence of this debate is: If you zoom into a specific unit (here coastal profile) of a global model, the results can differ from those of local analysis. However, on broader scales these differences tend to average out as local effects are not as dominant.

Irrespective of this debate, the reviewer's statement regarding our working assumption (i.e., "continuously increasing profiles") being wrong is not supported by the data. We have explicitly addressed this point in the manuscript and our results show that this is a perfectly valid assumption, with respect to the resolution of the elevation data. We must also note that the caption to Figure 1 explicitly refers to the "coastal profile" and not to the "beach profile", hence defining the scale of interest. Indeed, areas potentially flooded can extend several hundreds of km further inland than suggested by the reviewer, again making clear the potential scale of the features under investigation. Accordingly, the comment regarding coastal characteristics not being resolved in a 100m (or higher) resolution DEM is directly related to the scale of the study – this is the highest resolution that has been used in global studies and that is also used in some of the studies cited by the reviewer (e.g. Ramirez et al. 2016), where it is termed as hyper-resolution). The reviewer is thinking about processes which currently cannot be resolved at global scales as it is simply not possible (computationally) to use data of much higher resolutions, such as those employed in the local studies, e.g. Breilh et al. (2016), that the reviewer cites. More importantly, such data for coasts currently do not exist at global scale!

[Figure]

"At this stage it is important to highlight that the way the authors deal with coastal flooding is perfectly identical in terms of implementation to a case of inland flooding. The authors just use a DEM and forcing water levels along the coast which is identical to what the inland flooding community does with dynamic models long time ago. There are quite a few global river flooding impact assessments which use dynamic models and demonstrate that the time to leave the bathtub approach has arrived long time ago (Alfieri et al., 2017 and references therein; Winsemius et al., 2016)."

We fully agree with the reviewer regarding his comments on global river flooding – however, we believe that it is not good practice to simply equate coastal flooding with river/inland flooding. At the coast we are able to bring to the analysis a collation of observations (Table 1) which are specific to coastal wetlands. The significance of these observations are discussed in more detail below. The representation of processes that the reviewer refers to is currently the reality of global studies on coastal impact assessment. Lack of consistent global-scale coastal data and processing limitations when conducting the large number of model runs required for impact assessment (due to the different scenarios and time steps) have not yet allowed for additional complexity and there are currently no other global studies accounting for further processes (this is exactly what we are trying to do in this paper). We are also currently working on those aspects and hope to be able to address some of the above points in the near future.

The reasons for this gap are discussed in the manuscript (but also in other published work, e.g. Vafeidis et al., 2008; Hinkel et al., 2014) and are related to the lack of consistent data at global scale and to the computational costs for conducting the large number of model runs (due to the different scenarios and time steps) that are required for comprehensive impact assessments. Some of the studies that the reviewer cites (e.g. Seenath et al. 2016) also agree on this point and do not dismiss the bathtub method, pointing out that it can provide good results for some locations. Further, Seenath et al. correctly mention that hydrodynamic models also face several limitations when applied at broader scales.

"The authors resolve to searching the literature for reported water level reduction rates while the answer is in the basic textbooks. Flooding intensity is attenuated by bottom friction and the community has been estimating Manning friction coefficients depending on land use class for decades. There are also text book paper examples on how to relate the land use classes and the water level reduction they drive."

Our literature search aims to inform our sensitivity analysis. We are of course aware of Manning friction coefficients, which have been widely used to characterise these processes, although the range of values of Manning's 'n' (0.035 – 0.15 for floodplains) serves as an indication of the uncertainties involved. However, using such coefficients in a global study is far from trivial due to a range of issues, such as: the availability of consistent high-resolution global land use data, particularly for coastal areas where land-use maps show the highest errors; the empirical nature of the coefficient values, which greatly inhibits its use for global scale studies; the lack of information/values on different types of surfaces (e.g. urban surfaces at settlement or city scale) at global scale. Finally recent studies clearly show that water attenuation can have significant spatial variation even along the same land-use type (e.g. Stark et al. 2015). More fundamentally, the Manning's n approach may not be the most appropriate method to account for energy and momentum losses due to vegetated features. This is because the actual resistance to flow is not only through bottom friction but also from the form drag of plant architecture (stems, branches and leaves), particularly when the vegetation is emergent. Furthermore, there is the additional effect of reduction of surface winds in the presence of vegetation. Thus Table 1 presents a wholly new compilation not reported in any basic textbook, as far as we are aware. We report field studies of actual water level reduction rates, alongside numerical modelling efforts, to strengthen our analysis. Our study exactly emphasises in a quantitative manner the need of better understanding on how different types of land use attenuate water levels.

"The approach is outdated, the paper brings no new knowledge and there is no validation to support any claims. The authors should at least demonstrate that their approach

is valid based on some kind of validation against observations. They could also bring new knowledge to the community by recommending testing such an approach and testing against others, highlighting in which cases it could be valid, proposing calibration parameters etc"

We have addressed these points in our responses above and we believe that our findings are new and relevant for those working in the field of global coastal impacts and vulnerability. The reviewer may not be aware that for coastal sites at a global scale there is limited information available. Regarding the issue of validation we would like to repeat that our study presents a global sensitivity analysis (which is a form of validation); is based on a well published and evaluated global modelling framework; and is limited by validation issues that are common to global assessments that involve future projections. Finally we would like to stress that the main aim of the paper is to quantify uncertainty in global coastal impact assessments due to not accounting for water level attenuation. Our results show that uncertainties in impact assessment resulting from this omission are of a similar range to uncertainties related to total sea-level rise and provide an assessment of the large spatial and temporal variations that may result. For this purpose, our methods can provide important results to inform future adaptation policies at regional to global scales. It is still a question to be answered whether and to what extent a better representation of the physics would provide significant further insights in this context, particularly when considering the data limitations at global scale, and we are currently conducting research to explore this question.

References

Breilh, J.F., Chaumillon, E., Bertin, X. and Gravelle, M., 2013. Assessment of static flood modeling techniques: application to contrasting marshes flooded during Xynthia (western France). Natural Hazards and Earth System Sciences, 13(6), pp.1595-1612. Hinkel, J., Lincke, D., Vafeidis, A.T., Perrette, M., Nicholls, R.J., Tol, R.S., Marzeion, B., Fettweis, X., Ionescu, C. and Levermann, A., 2014. Coastal flood damage and adaptation costs under 21st century sea-level rise. Proceedings of the National Academy

of Sciences, 111(9), pp.3292-3297. Ramirez, J.A., Lichter, M., Coulthard, T.J. and Skinner, C., 2016. Hyper-resolution mapping of regional storm surge and tide flooding: comparison of static and dynamic models. Natural Hazards, 82(1), pp.571-590. Seenath, A., Wilson, M. and Miller, K., 2016. Hydrodynamic versus GIS modelling for coastal flood vulnerability assessment: Which is better for guiding coastal management?. Ocean & Coastal Management, 120, pp.99-109. Stark, J., Oyen, T., Meire, P. and Temmerman, S., 2015. Observations of tidal and storm surge attenuation in a large tidal marsh. Limnology and Oceanography, 60(4), pp.1371-1381. Vafeidis, A.T., Nicholls, R.J., McFadden, L., Tol, R.S., Hinkel, J., Spencer, T., Grashoff, P.S., Boot, G. and Klein, R.J., 2008. A new global coastal database for impact and vulnerability analysis to sea-level rise. Journal of Coastal Research, pp.917-924.

---

## Referee Comment (RC2) · Anonymous Referee #2 · 19 Oct 2017

I totally agree with referee #1, who has already expressed clearly and with good arguments what were my thought after reviewing this manuscript. The paper is really well written, what makes it easy to read, and the topic is of interest to a broader community. However, it is simplistic in the analysis, and the results don't give any new insights to the current state of the art. The authors use the flood module of DIVA, a large-scale coastal impact assessment tool to estimate the reduction on studied impacts (area and population exposure, and damages) for different water level attenuation rates on a global scale (based on a literature review), which are considered homogeneous and not dependent on the actual land use. Based on their results they highlight the importance of accounting for the effects of hydrodynamic processes on global scale analysis,

something that it is already known for the community (although avoided due to computational constrains). However, it is lacking for any validation data, and it also dismiss other sources of uncertainty such as the one introduced by a coarse digital terrain model (DTM), the damage functions, etc. (Moel et al. 2011). I believe this work could be a great base to develop new methodologies for global scale coastal flooding assessments, however I doubt if it brings new contribution to the community knowledge in its current state.

References Moel H., Aerts, J.C.J.H. (2011). Effect of uncertainty in land use, damage models and inundation depth on flood damage estimates.

---

## Referee Comment (RC3) · Anonymous Referee #1 · 13 Nov 2017

Having received the last response from the authors I would like to make my final comments in the interactive discussion..

- Our paper does not simply "highlight the importance of accounting for the effects of water level attenuation" but actually quantifies the uncertainty related to these effects in global impact assessments. Importantly it shows that differences resulting from the use of different RCPs are of the same magnitude as differences resulting from the lack of accounting for water-level attenuation.

Reviewer: This is partially true. To quantify the importance of water level attenuation one would need to have a proper estimate of water level attenuation. Instead the

authors use results from the 'tilted bathtub' (as some like to call it) but this by far over-simplifies most of the processes related to flooding. So this point is not valid, as the study is comparing two things that are both wrong.

- As the reviewers mention, the need to account for such effects is well known to the community but not implemented due to computational constraints. We agree and would also like to add data constraints to those limitations, which are at least equally important and which we now discuss in detail in the manuscript.

Reviewer: Fair enough..

- Our paper is the first study that addresses exactly these limitations for global coastal impact assessments under a full range of sea-level and socio-economic uncertainty. The study by Vousdoukas et al., (2016) is not global (contrary to what the reviewer suggests) and only assessed exposure for a limited set of physical scenarios (no socioeconomic) and a single event. - The main reason why no other study to date has carried out a global impact assessment using a hydrodynamic model is, besides the input data volume, the large number of runs that one would need to conduct for the analysis. Using a hydrodynamic model in our study would have involved at least 60 runs (20 time steps for three physical scenarios), with the number increasing greatly when additional scenarios (including socio-economic ones) are considered. The reviewers might be unaware of the above data and processing limitations for impact assessments when suggesting the use of hydrodynamic models (implemented e.g. through the use of specific Manning values). - We have conducted a sensitivity analysis (which is a form of validation), exactly because our study is global and involves future projections for a range of scenarios. - Including land use information (that both reviewers suggest or imply) in our assessment (e.g. for deriving Manning values) is not possible as there are currently no global consistent land-use scenarios available. Contrary to what the reviewers suggest, (as we discussed in our response to the first reviewer), such data are not readily available. Even existing current global land-use data (not scenarios) suffer from severe limitations (e.g. resolution, relevant land use classes) when it comes to

representing narrow coastal strips. On a more general note, we would like to point out that the use of hydrodynamic models does not always guarantee improved results, as explicitly stated by Vousdoukas et al. (2016). The bathtub model has so far been the only option for global impact assessment as it can produce reasonable results for many locations and under specific conditions (as clearly stated in all the references cited by the first reviewer); and is representative of maximum potential impacts. Importantly, it is questionable whether a more detailed representation of the physical processes (in this case flooding) is the most urgent requirement of impact assessments as there are many sources of uncertainties (data, physical scenarios, socio-economic scenarios, damage estimations) that may as well be of equal importance. Most of the papers that have been cited by the reviewers support this argument by underlining the above uncertainties. Our work is a first step towards developing a simple model for assessing flooding at global scales, by quantifying one important source of uncertainty. Importantly it suggests the possibility of emulating physical processes through the application of simple correction factors and a typological approach.

Reviewer: In all these arguments there are several statements which are either partially true, or partially relevant. For example, there are global land use datasets. They may have issues with accuracy and resolution but claiming that using a fixed coefficient for water level reduction is better than considering land use maps is at least a strange argument. Both reviewers do not question that the study is the first global assessment of the kind, we just both doubt about the validity of the findings. Any implementation of bathtub (tilted or not) or other simplified model on the existing global DEMs can be sufficient to address relative changes in future flooded areas, affected people, or damages. But the applicability of the method kind of stops there. Trying to quantify the uncertainty comparing to shaky flood models and without any validation is far fetched. Especially because the community has showed that we can approximate better the problem of coastal flooding. Sensitivity analysis is not validation. No other study has carried out a global COASTAL impact assessment using a hydrodynamic model or at least a more complex method, but there are several ones on river flooding. This goes

back to my previous comment on which the authors gave an incomplete reply. For sure coastal flooding is different than river one. In my opinion the benchmark is set by the studies of Bertin 2014, or McCall 2010, but these cannot be applied in large areas. Still anything beyond any bathtub solution would be a step forward. If the river community does it at global scale (CaMaFlood, GLOFRIS, GLOFAS) how can the authors claim it is not possible for coastal?

---

## Short Comment (SC1) · 13 Nov 2017

This is a personal short comment (not necessarily reflecting the views of my co-authors) to the last point raised by reviewer #1, suggesting the use of land-use data rather than a "tilted" bathtub with fixed coefficients; and regarding the lack of validation.

As we have mentioned in our previous response (and added to our manuscript), our study involves the use of future socio-economic scenarios (Shared Socioeconomic Pathways, SSP); unfortunately there are currently no consistent future land-use projections for these scenarios, thus not allowing for the consideration of land-use information (in the form of some type coefficients) in the assessment of impacts.

[Figure]

Regarding validation: we are reporting annual expected impacts (e.g. damages). To validate those impacts we would need long-term information on global impacts of coastal flooding. Such information is also generally not available, and where it is records are short and fragmented. Further, validation of flood extent by comparing to single events has been done in previous studies for few cases where some data have been available (also for the bathtub method which is well validated and its limitations are well known), and for single events (e.g. Xynthia). Such validation is not necessarily appropriate or informative in our case as flood characteristics can vary substantially based on storm characteristics (such as duration, wind direction etc.). Importantly, as our study also finds, flood characteristics are not necessarily the largest uncertainty in impact assessments.

The above are also two of the main reasons why we have conducted a sensitivity analysis (which is a form of validation according to all textbooks), which has led us to suggest other possible types of solutions for the representation of flood characteristics.

Finally, I would like to add that, personally, I have appreciated the discussion with the reviewer. I believe that such discussions are beneficial to all parts (even when there is disagreement) and promote science in general.

---

## Author Comment (AC2) · 13 Nov 2017

We note the brief comment on our paper by a second anonymous reviewer. Based on the comments of both reviewers we realise that many readers may not be familiar with global coastal impact assessments. Therefore we have reworded and extended sections of the document to clarify those points that appear to have led to confusion. Our reply to the first reviewer does actually address the comments of the second reviewer. However we would also like to draw attention to the following important points, which we have also now explicitly addressed in our manuscript:

- Our paper does not simply "highlight the importance of accounting for the effects of

water level attenuation" but actually quantifies the uncertainty related to these effects in global impact assessments. Importantly it shows that differences resulting from the use of different RCPs are of the same magnitude as differences resulting from the lack of accounting for water-level attenuation.

- As the reviewers mention, the need to account for such effects is well known to the community but not implemented due to computational constraints. We agree and would also like to add data constraints to those limitations, which are at least equally important and which we now discuss in detail in the manuscript.

- Our paper is the first study that addresses exactly these limitations for global coastal impact assessments under a full range of sea-level and socio-economic uncertainty. The study by Vousdoukas et al., (2016) is not global (contrary to what the reviewer suggests) and only assessed exposure for a limited set of physical scenarios (no socio-economic) and a single event.

- The main reason why no other study to date has carried out a global impact assessment using a hydrodynamic model is, besides the input data volume, the large number of runs that one would need to conduct for the analysis. Using a hydrodynamic model in our study would have involved at least 60 runs (20 time steps for three physical scenarios), with the number increasing greatly when additional scenarios (including socio-economic ones) are considered. The reviewers might be unaware of the above data and processing limitations for impact assessments when suggesting the use of hydrodynamic models (implemented e.g. through the use of specific Manning values).

- We have conducted a sensitivity analysis (which is a form of validation), exactly because our study is global and involves future projections for a range of scenarios.

- Including land use information (that both reviewers suggest or imply) in our assessment (e.g. for deriving Manning values) is not possible as there are currently no global consistent land-use scenarios available. Contrary to what the reviewers suggest, (as we discussed in our response to the first reviewer), such data are not readily available.

Interactive
comment

[Figure]

Even existing current global land-use data (not scenarios) suffer from severe limitations (e.g. resolution, relevant land use classes) when it comes to representing narrow coastal strips.

On a more general note, we would like to point out that the use of hydrodynamic models does not always guarantee improved results, as explicitly stated by Vousdoukas et al. (2016). The bathtub model has so far been the only option for global impact assessment as it can produce reasonable results for many locations and under specific conditions (as clearly stated in all the references cited by the first reviewer); and is representative of maximum potential impacts. Importantly, it is questionable whether a more detailed representation of the physical processes (in this case flooding) is the most urgent requirement of impact assessments as there are many sources of uncertainties (data, physical scenarios, socio-economic scenarios, damage estimations) that may as well be of equal importance. Most of the papers that have been cited by the reviewers support this argument by underlining the above uncertainties. Our work is a first step towards developing a simple model for assessing flooding at global scales, by quantifying one important source of uncertainty. Importantly it suggests the possibility of emulating physical processes through the application of simple correction factors and a typological approach.

---

## Referee Comment (RC4) · Anonymous Referee #2 · 14 Nov 2017

I have just read carefully the authors and referee #1 answers, and I must say that, once again, I do completely agree with referee #1. The idea presented in this work is relevant, and accepted. We all agree that the water extent experienced in a coastal flooding event, for similar hydrologic and topographic characteristics, is conditioned by the land use. However, the analysis performed is still far from innovative or scientifically relevant. The authors are not providing any solution to include land uses (or water level attenuations) in global coastal flood risk assessments. I believe that, it would be more relevant, to improve the actual land use databases, apply bathtub with a different coefficient of reduction associated to each land use, and compare it with the no-reduction case. Some data to validate would be crucial at this stage. And then, the same analysis

for different sea level rise scenarios could be performed. But at least, better capturing the spatial variability.